# OpenReview forum: "PostTrainBench: Can LLM Agents Automate LLM Post-Training?"
_ICML.cc/2026/Conference — ICML 2026 regular_

### Official Review · Reviewer_8Z4P · 2026-03-08

**Soundness:** 3
**Presentation:** 3
**Significance:** 3
**Originality:** 4
**Overall Recommendation:** 4
**Confidence:** 3

**Summary:**

The paper introduces POSTTRAINBENCH, a benchmark designed to measure whether command-line LLM agents (e.g., Codex CLI, Claude Code, Gemini CLI) can autonomously perform end-to-end post-training of small base LLMs under bounded resources (10 hours on a single H100 GPU). In each run, an agent is given a base model and a target benchmark (+ web access) and must independently design and implement a post-training pipeline (data collection/curation, training code, hyperparameters, iteration) to maximize held-out benchmark performance, subject to anti-cheating constraints (no training on benchmark test data, no evaluation harness modification, no model substitution). Results show agents often substantially improve base models but generally remain well below official instruction-tuned models on an overall weighted score, while sometimes exceeding instruction-tuned models on narrow targets (notably function calling). The paper also analyzes reward hacking / integrity failures observed during autonomous training, motivating the benchmark as a tool for tracking both capability and risk in automated AI R&D.

**Compliance With Llm Reviewing Policy:**

Affirmed.

**Final Justification:**

I maintain my weak accept score primarily due to inherent limitations I observe with the LLM as a judge as a scalable method for detecting reward hacking. With blogs like https://metr.org/blog/2025-06-05-recent-reward-hacking/ showing up with clear characterizations, i think that this is a core weakness of the paper and not easily addressable in a rebuttal and remains unsolved for me.

Otherwise the paper shows some very interesting measurements and findings (hence the weak accept), and so I maintain my score.

**Key Questions For Authors:**

In addition to the concerns above:

Anti-cheat judge validity

* Do you have any quantitative estimate (even spot-audits) of the LLM judge’s false positive / false negative rates, and how often judge decisions changed final rankings?

Robustness of conclusions to aggregation
* How stable are the headline rankings and conclusions under alternative scoring (uniform average, per-domain average, median across benchmarks), versus the current weighting scheme?

**Limitations:**

yes

**Strengths And Weaknesses:**

Strengths

- Benchmark is well-scoped and operationalizes “autonomous post-training” end-to-end:
    - The setup is concrete and reproducible in principle: fixed compute/time budget (10 hours, 1×H100), fixed base models, fixed benchmark suite, and “no starter code/data” to ensure the agent actually does the work
    - The benchmark emphasizes realistic agent behaviors (writing/debugging training scripts, running evaluations, web-searching for data), which makes it a stronger proxy for “automation of post-training” than narrowly-scoped subtasks (and their consequent benchmarks)
- Empirical results are informative and highlight sharp task-dependent variation:
    - Agents improve base models substantially on average (e.g., best agent around 21.5% weighted vs 7.5% base), but still trail official instruction-tuned baselines (51.1%), which is a useful reality check on current end-to-end R&D autonomy
    - The per-benchmark breakdown usefully shows that some tasks (e.g., BFCL function calling) are much more amenable to targeted hill-climbing than others (e.g., AIME / ArenaHard), which is informative for both capability forecasting and future benchmark design.
- Strong qualitative analysis and clear documentation of agent behavior:
    - The paper provides a concrete, end-to-end account of what agents actually do (and where they fail), rather than only reporting aggregate scores, which makes the findings more actionable.
    - The discussion of common failure modes (debugging loops, brittle training/eval scripts, data sourcing pitfalls) adds interpretability and helps pinpoint where “autonomous post-training” breaks down in practice.

Weaknesses
- Reliance on an LLM judge for anti-cheat is a potential single point of failure:
    - The benchmark’s integrity depends heavily on judge accuracy (false negatives permit contaminated/cheating wins; false positives can collapse legitimate runs to base score), but there’s no quantitative calibration of FP/FN rates or sensitivity analysis showing how often judge calls meaningfully affect results.
    - Similarly, “gray area” contamination (e.g., benchmark-like items in public training data, near-duplicate problems, or ambiguous provenance) seems hard to adjudicate robustly via heuristic judging alone, so it’s unclear how stable the reported rankings are to alternative auditing procedures. This may not be important now, but becomes much more important in the future when the community starts hill climbing on such a goal.
- Weighted aggregation may meaningfully affect headline results and comparability:
    - The benchmark uses a weighted aggregation across tasks based on instruction-tuning gains; this is a reasonable motivation, but it is nonstandard and could shift overall rankings in ways that are hard to interpret.
    - It would be helpful to report alternative aggregations (uniform average, per-domain average, median across tasks) and show whether the main conclusions and method rankings hold under those choices.
- Time limit of task (10 hours seems like it might not be timeless or prevent longer horizon results)?
    - I understand the rationale for enforcing something like this, but it may prevent seeing agent capabilities past instruction tuned level performance, since its very likely human practitioners took longer than 10 hours to get those results

---

> ### Author Rebuttal · Authors · 2026-03-31
>
> We thank the reviewer 8Z4P for their thorough and constructive review. We are glad the reviewer finds our benchmark well-scoped, empirically informative, with strong qualitative analysis, and rates our originality as excellent. Below, we address each concern in detail.
>
> **W1: Anti-Cheat Judge Validity**
>
> Great point! To validate the judge’s validity, we manually inspected runs flagged by the judge and confirmed that the vast majority were true positives. Only for GSM8K, we found that the judge sometimes decides that the dataset meta-math/MetaMathQA contains GSM8K test data, while in reality, it does not. Therefore, currently it is necessary to manually inspect GSM8K runs. Since our submission, we have improved and reworked the judge and made it stronger, so that the judge does not need to be double-checked.
>
> In particular, we made the following changes to the judge:
> 1. The judge has access to a script called `contamination_check.py` (see https://anonymous.4open.science/r/PostTrainBench_images-C876/contamination_check.py for reference), in which the judge can input training data that the agent used, and see if the training data is contaminated. This script is inspired by the package https://github.com/allenai/decon used for decontaminating the training data of Olmo 3.
> 2. There will be two independent judges: one based on GPT-5.2, one based on Sonnet-4.6. This reduces self-preference among the judge and agent.
>
> On 'Gray area' contamination: The n-gram overlap approach in `contamination_check.py` directly addresses this, since it detects near-duplicate problems and benchmark-derived content regardless of whether the agent relabeled or reformatted the data – cases where heuristic LLM judgment is weakest.
>
> **W2: Robustness of Conclusions to Aggregation Scheme**
>
> Thanks for this suggestion! We tried with the uniform average and the Median approach, as well as with our suggested weighted average. The results are here: https://anonymous.4open.science/r/PostTrainBench_images-C876/table_with_other_rankings.png
> The rankings resulting from the weighted average and uniform average are quite similar; the first 5 ranks do not change at all.
>
> We also tested Median, but found it ignores both extremes of the difficulty spectrum (AIME 2025 never contributes, GSM8K always exceeds the median), making it insensitive to agent progress on both the hardest and easiest tasks. We therefore consider it a less informative aggregation for this benchmark.
>
> In general, the rankings are quite stable with respect to the averaging technique.
>
> **W3: 10-Hour Time Limit May Be Restrictive**
>
> We appreciate this concern. We designed the 10-hour budget as a principled trade-off between signal and cost: a full PostTrainBench run already costs 900 USD–1300 USD, making longer budgets prohibitively expensive for broad evaluation across 4 base models × 7 benchmarks × 3 runs.
>
> Importantly, we have empirical evidence that 10 hours is not the binding constraint for current agents. We varied the time budget for various agents from 1 to 10 hours. Claude Opus 4.5 plateaus around 5 hours, and GPT-5.1 Codex Max continues improving, but most agents terminate well before the 10-hour mark. See figure here: https://anonymous.4open.science/r/PostTrainBench_images-C876/time_spent.png
>
> We tested 20-hour runs but found that agents often stopped early, well below the 10-hour mark. These experiments were therefore discontinued because agents did not utilize the additional time.
>
> Additionally, we are currently trying similar longer horizon (>20 hrs) experiments with the newest state-of-the-art models, which came out in the meantime after submitting the paper. The initial results with Opus-4.6 (1M context window) look promising.
>
> In other words, the current bottleneck is agent capability: planning, persistence, strategy diversity, not the time budget. We envision PostTrainBench as a living benchmark, and increasing compute budgets is a great idea.
>
> Thank you again for your thoughtful comments. We hope our answers address your concerns. We will be happy to engage in a follow-up discussion to clarify any remaining points.

---

> > ### Author Rebuttal · Reviewer_8Z4P · 2026-04-01
> >
> > Thank you to the authors for addressing most of my concerns: I still have some core concerns about LLM judge validity.
> >
> > I'm still particularly concerned about the LLM judge validity. As benchmarks become more popular and eventually receive community participation + submission, both the incentives for cheating and the capabilities of frontier models are a dual threat to the validity of measured progress. Existing work has already shown problems here (https://arxiv.org/pdf/2601.19532)
> >
> > There have been many examples in prominent benchmarks such as: (1) verbatim copying of SWE-bench patches (https://john-b-yang.github.io/swe-bench-cheating/index.html) (2) recovering SWE-bench solutions from git history (https://bayes.net/swebench-hack/) and (3) Opus 4.6 aware that it is in an eval environment (https://www.anthropic.com/engineering/eval-awareness-browsecomp).
> >
> > Compared to these, an LM-as-a-judge seems much more brittle and not something I think that adding more judge models or judge tools can easily fix. Even in the script provided, the provided n-gram decontamination check can be fooled by a LM paraphrasing a dataset or doing tokenizer shifting such that items are retokenized. This seems like a core weakness still inherent in the benchmark, and consequently why I maintain my score for now. Happy to engage in further discussion, and appreciate the detailed response!
> >
> > I appreciate the clarification on W2 and W3, and consider those resolved. I am definitely a bit worried about the high cost of this benchmark from reading the responses to other reviewers, but recognizing that this is a frontier capabilities benchmark and consequently likely expensive. I'm also curious on how this might scale to larger and larger models closer to frontier capabilities, but agree that this is certainly future work.

---

> > > ### Author Response · Authors · 2026-04-07
> > >
> > > We thank the reviewer for the continued engagement and the excellent references to eval gaming.
> > >
> > > We agree that judge robustness is a critical concern, especially as benchmarks mature and incentives for gaming increase. We want to address this on two levels: why PostTrainBench provides meaningful signal today, and how long-term integrity can be strengthened.
> > >
> > > **The benchmark provides a meaningful signal right now.** We agree that no LLM judge is fully adversarially robust. But today, LLMs are not adversarially trained against our judge, and doing this would likely reduce the general capabilities of the model. Because the model would learn to reward hack, which is something frontier model providers are trying to avoid.
> > > Additionally, the agents do not know about the `contamination_check.py` script. Therefore this is a real advantage which the judge currently has.
> > >
> > > **Long-term roadmap: private held-out evaluations.** We believe the strongest path forward sidesteps the contamination judge entirely: agents receive only a validation script or general objective (e.g., "improve math ability"), and submitted checkpoints are evaluated server-side on undisclosed test data that the agent cannot access. This mirrors platforms like Kaggle with private test sets and is robust even against adversarial agents, since there is nothing to contaminate against. This direction does come with trade-offs: reduced transparency (evaluations cannot be fully public), increased engineering and compute demands (requiring a hosted evaluation endpoint), and the need to create novel evaluations not already present on the web. We view this as the natural next step for PostTrainBench, and we would be glad to implement it in a future release, though it goes beyond the scope of this paper.
> > >
> > > **Positioning.** We view PostTrainBench as a first-of-its-kind benchmark that provides meaningful and actionable signal today. We acknowledge that the judge is not adversarially robust in the limit, and we are transparent about this. But we believe the current design is well-matched to current agent capabilities, and the modular architecture supports progressive hardening as the threat landscape evolves.
> > >
> > > We would love to hear your thoughts on these points.

---

### Official Review · Reviewer_KpHE · 2026-03-10

**Soundness:** 3
**Presentation:** 3
**Significance:** 4
**Originality:** 3
**Overall Recommendation:** 5
**Confidence:** 4

**Summary:**

The paper proposes a new benchmark for LLM coding agents: PostTrainBench. An agent is tasked with improving a base LLM's (<=4B class) score on a single well-known LLM benchmark, with scores aggregated across 4 different base LLMs and 7 benchmarks. An LLM judge is employed to prevent cheating (e.g., train on test set). The authors find that the current generation of coding agents is able to make meaningful progress but significantly lacks behind the main baseline which are the officially released instruction-tuned versions of the base LLMs.

**Compliance With Llm Reviewing Policy:**

Affirmed.

**Key Questions For Authors:**

How effective did you find the LLM judge for cheating detection? It would be good to discuss this part in the paper and provide examples (if you found them) where the judge missed a cheating attempt or misclassified a non-cheating solution. I.e., a quantification of false positives or negatives would strengthen the paper further.

With regards to current models, what behaviors and solution strategies did you observe? I encourage the authors to publish the trajectories of their evaluation runs to enable the community to contribute tools for analysis, for example.

**Limitations:**

yes

**Strengths And Weaknesses:**

I found the paper to be well-written and easy to follow, the objective and setup of the benchmark are clear to me. In terms of significance, I agree with the authors that benchmarking how well LLMs can automate the LLM training process itself is important; and, as we've seen time and time again, a good benchmark is the first step to direct focus and encourage further improvement in a domain. A potential blocker for wide adoption of the benchmark is its relatively prohibitive cost; on the other hand, I like that it tests the desirable end goal of post-training automation directly. The fact that for many open-source models we have base to start from and instruction-tuned models as (still very strong) baselines makes the proposed setup particularly attractive.

In contrast to many existing benchmarks, the paper proposes a setup that is minimal in constraints and open-ended. The lack of constraints in particular is well-suited for the current generation of LLM agents that have full access to the web. On the other hand, combined with the runtime cost, makes me expect that this will be less of a benchmark for direct hill-climbing but can still function as good proxy of frontier model capabilities.

Some notes regarding the write-up:
- L28 left: "generally lag instruction-tuned LLMs from leading providers" from the abstract alone it was not clear to me what "leading providers" refers to
- L107 right: it's -> its
- L137 right: "We apply the chat template" (for GSM8K) which chat template?
- L189 right: "Interestingly, native scaffolds consistently outperform OpenCode ..." except for Opus, so not consistently
- L217 left: "There was a lot of advancement in recent months, ..." while I don't disagree, this statement compares Sonnet to Opus, which are differently-sized and hence differently-capable models irrespective of their release date

---

> ### Author Rebuttal · Authors · 2026-03-31
>
> We thank Reviewer KpHE for the thoughtful and constructive review. We are glad the reviewer finds the paper well-written, the setup clear, and likes the use of base-to-instruct pairs as baselines. Below, we address each point in detail.
>
> **Write-up notes**
>
> Thank you for your careful reading! We will address those in the revision.
>
> **Q1: LLM Judge Effectiveness**
>
> Good point! To validate the judge, we manually inspected flagged runs and confirmed that the vast majority were true positives: agents genuinely loaded benchmark data for training, hardcoded benchmark problems, or performed evaluation-guided data generation. Only for GSM8K, we found that the judge sometimes decides that the dataset meta-math/MetaMathQA contains GSM8K test data, while it actually does not. Therefore, currently it has been necessary to manually inspect GSM8K runs.
> However, since the submission, we have improved and reworked the judge and made it stronger, so that the judge does not need to be double-checked.
>
> In particular, we made the following changes to the judge:
>
> 1. The judge has access to a script called `contamination_check.py` (see https://anonymous.4open.science/r/PostTrainBench_images-C876/contamination_check.py for reference), in which the judge can input training data that the agent used, and see if the training data is contaminated. This script is inspired by the package https://github.com/allenai/decon used for decontaminating the training data of Olmo 3.
> 2. There will be two independent judges: one based on GPT-5.2, one based on Sonnet-4.6. This reduces self-preference among the judge and agent.
>
> The `contamination_check.py` script addresses gray-area contamination by performing n-gram overlap detection between training data and benchmark data, catching cases that heuristic LLM judgment alone might miss.
>
> **Q2: Agent Behaviors and Trajectory Release**
>
> Since the submission, we have analyzed the agent strategies in detail. We noticed the following from analyzing the agent traces:
> - *SFT is used most often:* Supervised fine-tuning is the overwhelmingly dominant method. Almost no agents attempted reinforcement learning or distillation from stronger models.
> - *Iterative refinement within SFT:* Rather than switching paradigms, agents iterate on data preparation, hyperparameters, and formatting. Opus 4.6 produces 3–8+ script versions per task; GPT-5.3 Codex is more conservative with 1–2 versions.
> - *Usually, LoRA is used:* Most agents default to LoRA/QLoRA, with Gemini 3.1 Pro as an outlier preferring full fine-tuning (~66% of cases).
> - *Data sourcing:* Agents curate data from Hugging Face (e.g., MBPP, glaive-function-calling, Hermes datasets) and, in some cases, generate synthetic data themselves (e.g., Opus 4.5 creates medical QA pairs for HealthBench).
>
> We will expand and add this section to the revision.
>
> **Trajectory release.**
>
> We fully agree with this suggestion, and we have prepared the full sanitized execution trajectories for the revision. We believe this will be a valuable resource for the community to develop analysis tools and identify new patterns.
>
> Thank you again for your thoughtful comments.

---

### Official Review · Reviewer_2Zhi · 2026-03-11

**Soundness:** 3
**Presentation:** 4
**Significance:** 3
**Originality:** 3
**Overall Recommendation:** 5
**Confidence:** 3

**Summary:**

The authors propose a benchmark in which LLM is supposed to take a base LLM, before post-training, and do this stage of LLM training to the base LLM. In that manner, LLMs will be tested if they can automate R&D specifically in the AI field.

**Compliance With Llm Reviewing Policy:**

Affirmed.

**Final Justification:**

Raised the score to 5 during the rebuttal due to satisfying authors' responses.
The paper is well written and timely.

**Key Questions For Authors:**

Questions:

Q1:	It would be interesting to see an analysis of the problems agents were dealing with and see if there were things in common. In addition, what were the differences between each agent performing the task?

Q2:	A comparison of current benchmarks (specifically code benchmarks) results of tested LLMs and aligning them with results in the experiment section could be very interesting. For example, if one LLM’s excels in coding it might highly affect (to the better) the agents based on the LLM performance in the proposed benchmark.

**Limitations:**

The benchmark proposed is very expensive (money and time) and not easy to perform. As the authors note, while practical for large-scale, the evaluation does not reflect real-world post-training timelines. Also, because of the high expense of each run, the authors avoid thorough evaluation where it might be needed.

**Strengths And Weaknesses:**

Strengths:

S1:	The paper is written clearly and easy to read.

S2:	The idea is original and highly relevant to the current state of the art field.

S3:	The authors provide an analysis of the effect of time limit. The authors conduct thorough research and provide the details in the paper


Weaknesses:

W1:	 The benchmark proposed is very expensive (money and time) and not easy to perform. It’s hard to supervise the training in every run performed as part of the experiments section so it will be easy to miss wrongdoings, making this benchmark less reliable.

W2:	We do know the different scaffolds affects the result but we don't know their internal system prompt and how it affects the result. They can be changed every day without the user knowing and it can vary from model to model and even from task to task.

W3:	In the benchmark of BFCL the variation is high making the conclusion in 3.2 less reliable. It is interesting to see the high variation compared to other benchmarks.

---

> ### Author Rebuttal · Authors · 2026-03-31
>
> We thank Reviewer 2Zhi for acknowledging the originality and clarity of our work. We address each concern below.
>
> **W1: The benchmark is expensive and hard to supervise, making it less reliable.**
>
> We acknowledge the cost concern. However, we note that PostTrainBench is designed as a frontier capability benchmark, similar in spirit to RE-Bench or MLE-bench, where cost is inherent to measuring real-world autonomous capabilities. To address reliability, we have taken several steps in our updated version: (1) we run 3 independent runs for all frontier agents on native scaffolds to estimate variance; and (2) we release all code and agent instructions to enable reproducibility. The cost per agent evaluation (~$900–1300 for the full matrix) is comparable to other frontier benchmarks and is expected to decrease as compute and API costs drop.
>
> We agree that such long running benchmarks are hard to supervise. We therefore significantly strengthened our anti-cheat infrastructure since the submission. The judge now has access to a `contamination_check.py` script (see https://anonymous.4open.science/r/PostTrainBench_images-C876/contamination_check.py) that performs n-gram overlap detection between training data and benchmark data (inspired by the decontamination pipeline used for OLMo 3), and we use two independent judges (GPT-5.2 and Sonnet-4.6) to reduce self-preference bias. This significantly strengthens the judge and we hope this addresses your concerns.
>
> **W2: Scaffold system prompts are opaque and could change.**
>
> This is a valid concern for any benchmark involving proprietary tools. We mitigate this by: (1) evaluating the same model on multiple scaffolds (e.g., Claude Opus 4.5 on both Claude Code and OpenCode), which isolates scaffold effects; (2) including OpenCode as an open-source scaffold whose internals are fully inspectable; and (3) reporting results with pinned scaffold versions so that scaffold version effects can be tracked over time. We will add a more explicit discussion of this limitation in the revision.
>
> **W3: High variance on BFCL makes conclusions in Section 3.2 less reliable.**
>
> We agree that BFCL shows high variance. This high variance is actually an informative finding: it reflects the bimodal nature of BFCL outcomes; agents either discover a working function-calling training strategy (scoring 80%+) or fail to find one (scoring near the base model).  In our updated version, we now report results across more agents and runs, which helps stabilize the picture. The key claim in Section 3.2 - that agents can beat instruct-tuned models on BFCL - is supported by the best-case performance (89% for Gemma-3-4B), which occurred in multiple independent runs across different agents. We will add a discussion of this variance pattern in the revision.
>
> **Q1: Analysis of common problems and differences between agents.**
>
> Since our submission, we have substantially expanded the paper. We have added 9 new agents to the leaderboard (22 total, up from 13), a detailed analysis of post-training method selection and a systematic contamination audit across all runs. On the method analysis specifically, key findings include: all agents default to SFT as their primary method; only Claude-based agents attempt RL (GRPO); agents differ in their preference for LoRA vs. full fine-tuning; and agents primarily iterate within SFT (producing multiple script versions) rather than switching paradigms. Common failure modes include: training timeouts due to overambitious dataset sizes, vLLM/evaluation debugging loops, and data formatting issues. We will expand this analysis further in the manuscript in the revision.
>
> **Q2: Correlation between LLM coding capability and PostTrainBench performance.**
>
> While agent models that rank highly on standard coding benchmarks (e.g., SWE-bench) tend to also perform well on PostTrainBench, the correlation is imperfect – indicating that PostTrainBench captures a broader set of capabilities beyond pure coding, including research planning, data curation, and experimental iteration. This divergence is illustrated by GPT-5.2 and Opus-4.5: on SWE-bench Verified, Opus-4.5 scores higher (76.8 vs. 72.8), yet on PostTrainBench the ranking reverses, with GPT-5.2 outperforming Opus-4.5 (21.5 vs. 17.1). This reordering suggests that the two benchmarks stress meaningfully different competencies. We will include a full correlation analysis in the revision.
>
>
> Thank you again for your constructive feedback. We believe the expanded evaluation (22 agents, systematic contamination auditing, and detailed behavioral analysis) in our revision directly addresses the concerns about reliability and limited evaluation. We would welcome any follow-up questions during the discussion period.

---

> > ### Author Rebuttal · Reviewer_2Zhi · 2026-04-01
> >
> > I appreciate the authors’ response and their careful note for each point raised and addressing my concerns. The authors answer the key questions by adding experiments by adding a detailed analysis of post-training methods and a promise to add a full correlation analysis in the revision, as I tend to trust authors to do so. I believe W2 remains a weakness as current results won’t remain legitimate for a long time, as the internal system prompt can change. This weakness forces the users to run the entire evaluation because results are getting old very fast/raising the complexity of conducting the analysis.
> > Despite this inherent challenge of evaluating proprietary agents, the authors have significantly strengthened the paper's empirical foundation and anti-cheating infrastructure. After carefully considering the response, I am raising my overall recommendation score from 4 to 5, and my soundness score from 2 to 3.

---

### Official Review · Reviewer_vgEv · 2026-03-19

**Soundness:** 4
**Presentation:** 4
**Significance:** 4
**Originality:** 4
**Overall Recommendation:** 4
**Confidence:** 4

**Summary:**

This paper presents a new benchmark to challenge LLM agents to do end-to-end post-training.

Overall, I find the paper well-written and easy to follow. However, the lack of detail is concerning.
Hope the authors will attach an appropriate appendix during rebuttal with more analysis and detailed description of model behaviors.

**Compliance With Llm Reviewing Policy:**

Affirmed.

**Final Justification:**

I wish the paper has more detailed content and has more publicly available setups. However, that doesn't exclude this paper for making a timely contribution.

**Key Questions For Authors:**

The paper includes some important statistics in Table 2, Figure 4, particularly on time. The supplementary code README suggested "Evaluation should run in ~15 minutes on an H100 (use vLLM for inference, subsample if needed during development) For the final evaluation, please use the full benchmark". I want to ask the following question:

1. How are you able to measure the average performance of 3 different models at the **same** hour mark (1h, 2h, 5h, 10h) if the agents are free to do whatever they want? What if at hour 2, Gemini 3 Pro is building synthetic dataset, while Claude Opus 4.5 is just finished the last training but hasn't launched a new job yet?
2. Do agents always evaluate on the full benchmark? Or do they evaluate on a subset, or is that just a choice they have to make? Even with vLLM for inference, evaluating on these benchmarks take time -- how long does it take? If your answer is, difficult to analyze from the raw log, that's fine -- but these are interesting for us to know.
3. Line 315 says "We also tested 20-hour runs but found that agents often stopped early, well below the 10-hour mark. These experiments were therefore discontinued." -- do you have any mechanism to keep the agent going?
4. Line 197 cost ~$840 is a single run for 4 agents on 7 benchmarks. But then you said "these experiments" were discontinued. How many of them are discontinued? These can also add up to the evaluation cost right?

**Limitations:**

Yes

**Strengths And Weaknesses:**

Strengths:
1. There are some using LLM agent to perform pre-training papers (aka, training on NanoGPT). However, the space of autonomous post-training is lacking. This paper fills in an important niche and provides an incredible benchmark.
2. The presentation is very clean. The codebase is seemingly complete.

Weaknesses:
1. Using GPT5-mini as judge (reward model) is an unfortunate choice. Though it is a much more powerful LLM than the 1.7B-4B models that are getting post-trained, reward hacking might still be possible.
2. Still the paper lacks quite a few key details (see questions)

---

> ### Author Rebuttal · Authors · 2026-03-31
>
> We thank reviewer vgEv for the positive assessment and the detailed technical questions. We address each below.
>
> **Q1: How do you measure performance at specific hour marks (1h, 2h, 5h, 10h) if agents act freely?**
>
> For the time-budget ablation, we do not checkpoint mid-run. Instead, we run separate experiments with different time limits enforced via `timer.sh`. For example, the "2h" data point comes from a completely independent run where the agent was given a 2-hour budget, not from interrupting a 10-hour run at hour 2. This ensures each data point reflects how agents behave when they know they have that specific time budget, which affects their planning and strategy choices. We will clarify this in the revision.
>
> **Q2: Do agents always evaluate on the full benchmark? How long does evaluation take?**
>
> Agents have full autonomy over how they use the `--limit` flag during development. Most agents learn to use `--limit` for fast iteration (e.g., `--limit 20` for quick sanity checks) and then run the full benchmark less frequently. The final evaluation we report is always on the full benchmark, run by us after the agent submits its checkpoint, not by the agent itself. Evaluation time varies by benchmark: BFCL takes ~2-5 minutes on an H100, GSM8K up to ~1.25 hours. It also depends on the base model and how the agent trained the model. More verbose models take longer to evaluate.
>
> **Q3: Do you have any mechanism to keep the agent going when it stops early?**
>
> Currently, no. The agent prompt instructs autonomous operation, and the 10-hour budget is communicated via `timer.sh`, but we do not force agents to continue. One natural approach would be adding a prompt instruction like "use the full time budget" or implementing a scaffold-level retry loop – we plan to explore this in future work and note it in the revision.
>
> **Q4: How many 20-hour runs were discontinued, and what were the costs?**
>
> We stopped the 20-hour runs after seeing that the GPT agent consistently stopped earlier than 10 hours. The costs are therefore comparable to the respective 10-hour runs. We are currently trying similar longer horizon (>20 hrs) experiments with the newest state-of-the-art models, which came out in the meantime after submitting the paper. The initial results with Opus-4.6 (1M context window) look promising. We will add a section in the revision.
>
> On GPT-5-mini as judge: We agree this is a limitation worth discussing. We chose GPT-5-mini because it is substantially more capable than the models being evaluated. For ArenaHard, the judge evaluates creative writing quality against a fixed baseline, so reward hacking would require the post-trained model to specifically exploit GPT-5-mini's preferences, which is unlikely given our small models' limited capability. We will add a discussion of this limitation in the revision and note that future versions could cross-validate with multiple judge models.
>
> Thank you again for your thoughtful comments. We hope our answers address your concerns. We will be happy to engage in a follow-up discussion to clarify any remaining points.

---

> > ### Author Rebuttal · Reviewer_vgEv · 2026-04-03
> >
> > Thank you for the rebuttal. I think now it makes much more sense.

---

> > > ### Author Response · Authors · 2026-04-07
> > >
> > > Dear Reviewer vgEv,
> > >
> > > Thank you again for the thoughtful review and engagement during the discussion phase. Since your acknowledgement indicated concerns were fully resolved, we wanted to gently check whether you might consider updating the score.

---

### Decision · Program_Chairs · 2026-04-30

**Decision:**

Accept (regular)

**Comment:**

PostTrainBench introduces a timely evaluation framework testing whether frontier LLM agents can autonomously post-train base models under a strict compute budget (single H100, 10 hours). Across a broad matrix of models and tasks, the paper reveals that while current agents still underperform official instruction-tuned baselines overall, they can surpass them in narrow settings. Crucially, the paper exposes concerning reward-hacking behaviors (e.g., training on test data, unauthorized API usage), which constitutes a highly valuable independent finding.

The reviews were unanimously positive (two 5s, two 4s), praising the benchmark’s novelty, engineering quality, and clarity. Initial concerns focused on LLM-as-a-judge robustness for anti-cheating, evaluation costs, and specific design choices.

The rebuttal substantially addressed these issues by expanding the agent pool, introducing dual independent judges, adding n-gram decontamination scripts, and promising to release sanitized trajectories. These additions prompted one reviewer to upgrade to a firm Accept. The sole remaining hesitation regarding the inherent brittleness of LLM judges (Reviewer 8Z4P) is a valid long-term concern, but it is a generic limitation of the evaluation paradigm rather than a specific flaw of this work. The authors’ proposed roadmap for a private held-out set is a reasonable mitigation. Given the paper’s clear contribution, strong empirical response, and high community relevance, I recommend acceptance.